# One Stone, Three Birds: Feasible Tuning of Barrier Heights Induced by Hybridized Interface in Free-Standing PEDOT@Bi_2_Te_3_ Thermoelectric Films

**DOI:** 10.3390/polym16141979

**Published:** 2024-07-11

**Authors:** Li Feng, Fen Wang, Hongjie Luo, Yajuan Zhang, Jianfeng Zhu, Yi Qin

**Affiliations:** 1Shaanxi Key Laboratory of Green Preparation and Functionalization for Inorganic Materials, Institute of Silicate Cultural Heritage, School of Materials Science and Engineering, Shaanxi University of Science and Technology, Xi’an 710021, China; fengl@sust.edu.cn (L.F.); zyj08216023@126.com (Y.Z.); zhujf@sust.edu.cn (J.Z.); 2Research Institute of Cultural Relics, Shanghai University, Shanghai 200444, China; hongjieluo@shu.edu.cn

**Keywords:** free-standing film, PEDOT@Bi_2_Te_3_, defect level, Bi_2_Te_3_ nanosheets, conductivity

## Abstract

Converting low-grade thermal energy into electrical energy is crucial for the development of modern smart wearable energy technologies. The free-standing films of PEDOT@Bi_2_Te_3_ prepared by tape-casting hold promise for flexible thermoelectric technology in self-powered sensing applications. Bi_2_Te_3_ nanosheets fabricated by the solvothermal method are tightly connected with flat-arranged PEODT molecules, forming an S-Bi bonded interface in the composite materials, and the bandgap is reduced to 1.63 eV. Compared with the PEDOT film, the mobility and carrier concentration of the composite are significantly increased at room temperature, and the conductivity reaches 684 S/cm. Meanwhile, the carrier concentration decreased sharply at 360 K indicating the creation of defect energy levels during the interfacial reaction of the composites, which increased the Seebeck coefficient. The power factor was improved by 68.9% compared to PEDOT. In addition, the introduction of Bi_2_Te_3_ nanosheets generated defects and multidimensional interfaces in the composite film, which resulted in weak phonon scattering in the conducting polymer with interfacial scattering. The thermal conductivity of the film is decreased and the ZT value reaches 0.1. The composite film undergoes 1500 bending cycles with a 14% decrease in conductivity and has good flexibility. This self-supporting flexible thermoelectric composite film has provided a research basis for low-grade thermal energy applications.

## 1. Introduction

Thermoelectric material is an important functional material, it can be electrical energy and thermal energy conversion. Thermoelectric devices manufactured are energy-saving, non-polluting to the environment, have good stability and long service life, and many other advantages. Therefore, in the modern construction of environmentally friendly social applications, they are increasingly important. Harnessing waste heat to generate electrical energy is essential for the swift advancement of alternative energy technologies. Thermoelectric devices can directly transform waste heat into electrical energy [1,2,3,4]. The assessment of thermoelectric materials’ efficiency is conducted using the formula ZT = S^2^σT/κ. Seebeck coefficient (S) is a measure of the magnitude of the voltage induced by a temperature difference in a thermoelectric material. Conductivity (σ) is a parameter used to describe the ease of charge flow in a substance. Thermal conductivity (κ) is the amount of heat transferred per unit of time per unit of horizontal cross-sectional area. These three parameters interact with each other to affect the efficiency of thermoelectric conversion. The parameters S, σ, and κ are intricately interconnected with the electronic structure and charge carrier scattering, resulting in their interdependence [5,6]. Therefore, achieving high ZT values is challenging. Theoretical computations and experimental observations suggest that low-dimensional thermoelectric materials exhibit enhanced ZT values due to the influence of quantum size effects and the scattering of phonons at interfaces. Creating nanostructures and incorporating doping can increase the density of states near the Fermi level through the utilization of quantum confinement, making electron levels more conducive to movement within the material [7,8,9,10]. This enhances the electron transport properties, providing an effective pathway for decoupling electrical conductivity and thermal conductivity [11].

Inorganic semiconductors with high Seebeck coefficients have become the preferred choice for preparing composite flexible thermoelectric films with wearable potential. Based on this, bismuth telluride (Bi_2_Te_3_) with a high Seebeck coefficient is used to prepare composite materials [12,13]. Bi_2_Te_3_ is a better inorganic thermoelectric material with good utilization of low-grade thermal energy to generate electricity. Bi_2_Te_3_ belongs to the R-3m rhombohedral crystal system and is a semiconductor compound composed of elements from Group V and Group VI. It operates in the temperature span from 300 K to 450 K, featuring a narrow bandgap of only 0.15 eV, making it a typical narrow-bandgap semiconductor that facilitates electronic transitions [14,15,16,17]. As a crucial thermoelectric material of current interest in the low-temperature and room-temperature range, various morphologies of Bi_2_Te_3_-based thermoelectric materials have been prepared using physical and chemical methods, including nanowires, nanoplates, and nanoparticles [11,18,19,20,21].

The combination of Bi_2_Te_3_ with different morphologies and PEDOT has been gradually studied in the field of thermoelectric conversion, but the thermoelectric performance of self-supporting films needs to be improved [22,23]. Thongkham et al. [24] employed an in-situ synthesis method to create Bi_2_Te_3_-PEDEOT:PSS nanowire thermoelectric materials on Te nanowires. At a mass ratio of 1% for Bi_2_Te_3_, the power factor reaches 7.5 μW/mK^2^. Xiong et al. [25] produced AB and ABA-type films using a layer-by-layer coating method. The h-AB-type film with 10 wt.% Bi_2_Te_3_ NWs demonstrated the highest power factor of 10.6 μW/mK^2^. Wang et al. [26] utilized thermal evaporation deposition and nanoimprint lithography to create Bi_2_Te_3_ nanosphere arrays. In the interstitial spaces of the array, a gas-phase polymerized PEDOT continuous phase was applied. Despite a composite film thickness of only 100 nm, the power factor of Bi_2_Te_3_-PEDEOT:PSS achieved 1350 μW/mK^2^, with a ZT value of 0.56. This can be ascribed to the scattering of phonons at the interface of Bi_2_Te_3_ nanospheres within the film, while the continuous PEDOT ensures effective electron transport. Zheng et al. [27] synthesized Bi_2_Te_3_ nanowires dispersed in PEDEOT:PSS, spin-coated into a film, and the energy-filtering effect generated an effective improvement in the Seebeck coefficient, resulting in the achievement of a ZT value of 0.2 for the composite film. This indicates that the thermoelectric properties of PEDOT films can be enhanced by composite Bi_2_Te_3_. However, in recent years, most of the composite films studied were prepared on glass substrates or silicon wafers using spin-coating or vapor-phase deposition processes. The thickness of the film is less than 100 nm, which is brittle and not suitable for mass production and preparation. Thermoelectric conversion performance needs to be improved. Facing the actual demand of wearable applications, the thermoelectric and mechanical properties of these composite films still need to be improved. Meanwhile, regarding the emerging composite systems in two-dimensional configurations, previous studies have mainly focused on film preparation techniques and post-processing property enhancement, and their specific features in terms of the structural morphology of materials and carrier transport behaviors have not yet been comprehensively analyzed.

Herein, we successfully prepared Bi_2_Te_3_ nanosheets by solvothermal method, and used the spatial site resistance effect of polyvinylpyrrolidone (PVP) to modulate the morphological design of Bi_2_Te_3_. Free-standing PEDOT@Bi_2_Te_3_ composite films were prepared by combining PEDOT with them through an electrostatic self-assembly method and casting film-forming process. Composite films produced not only achieved a uniform distribution of Bi_2_Te_3_ in PEDOT but also integrated the high conductivity and a large Seebeck coefficient of the two components, resulting in improved thermoelectric properties of the composite films. Charge transfer mechanisms considering composition and orientation are proposed for this specific 2D nanosheet binary composite system. Direct interfacial effects of the materials are also discussed. With high thermoelectric properties and excellent flexibility, the PEDOT@Bi_2_Te_3_ composite films developed in this work have promising applications in wearable thermoelectric devices or sensor applications.

## 2. Materials and Methods

### 2.1. Synthesis of Bi_2_Te_3_ Nanosheets

Preparation of Bi_2_Te_3_ nanosheets using solvothermal method. All experimental reagents were purchased from Sinopharm, and no further purification was required. PVP (1 g) was placed in a beaker, and 50 mL of ethylene glycol was added. The mixture is vigorously stirred until a transparent solution is formed. According to the molar ratio, BiCl_3_ (0.6301 g), TeO_2_ (0.48 g), and NaOH (1 g) are added to the precursor. Stirring continues for 90 min until a clear and semi-transparent solution is obtained. The mixture is then poured into a reaction vessel lined with polytetrafluoroethylene and reacted at 180 °C in a convection oven for 9 h, followed by cooling to room temperature. Multiple centrifuge washes are conducted using acetone, anhydrous ethanol, and deionized water. The solid precipitate is collected, placed in a vacuum drying oven at 65 °C, and dried for 12 h, resulting in the black Bi_2_Te_3_ solid power. 

### 2.2. Preparation of PEDOT@Bi_2_Te_3_ Flexible Films

Different masses of monodisperse Bi_2_Te_3_ nanosheets were selected as inorganic fillers and added to a PEDOT:PSS-doped 10 vol.% DMSO solution. The resulting mixed solution was thoroughly stirred for 6 h at room temperature. Subsequently, it was cast onto the processed glass substrates and dried in a vacuum oven at 50 °C for 12 h. The film was soaked in water, peeled off from the glass substrate, and dried at 80 °C for 4 h to obtain a smooth-surfaced PEDOT@Bi_2_Te_3_ flexible composite film with a thickness of 10 μm. According to the experimental process, the schematic structure of PEDOT@Bi_2_Te_3_ composited film is depicted in Figure 1. Based on the amount of Bi_2_Te_3_ filler (5 wt.%, 10 wt.%, 15 wt.%, 20 wt.%, 25 wt.%), the obtained PEDOT@Bi_2_Te_3_ flexible composite thermoelectric films were labeled as PB-5, PB-10, PB-15, PB-20, and PB-25, respectively.

### 2.3. Characterization and Property Measurement

The surface and sectional morphology of Bi_2_Te_3_ powder and PEDOT@Bi_2_Te_3_ film were investigated by scanning electron microscopy (FESEM; Hitachi S4800, Tokyo, Japan). The Raman spectroscopy of PEDOT@Bi_2_Te_3_ film was acquired using Renishaw-InVia equipped with a 532 nm red laser and a CCD detector. The infrared spectra of the films were performed by a Fourier-transform infrared spectrometer (FT-IR; Bruker Vertex 70, Broken, Germany). The doping levels and element valence of the composite film were determined through X-ray photoelectron spectroscopy (XPS; Kratos Axis Supra, Manchester, UK), and XPS spectra were calibrated with the binding energy of the standard C 1s (284.8 eV). The characteristic parameters of the nanofilms, including conductivity, Seebeck coefficient, and power factor, were evaluated using a thermoelectric test system (CTA-3), and the conductivity and Seebeck coefficient were observed employing static direct current and four-terminal methods under a helium atmosphere. The flexibility and sensitivity of the composite films were measured by a Keithley 2400 SourceMeter instrument from the Mansfield, TX, USA. Carrier concentration and mobility inside the free-standing PEDOT@Bi_2_Te_3_ flexible thermoelectric films were characterized using the MMR K2000 controlled and continuously variable-temperature Hall-effect test system from the USA. The tested film was a 10 mm × 10 mm square with a thickness of 10 μm, and a magnetic field strength of 5000 G was applied.

## 3. Results and Discussion

### 3.1. Microstructure and Crystal Structure of Bi_2_Te_3_ Nanosheets

The successful synthesis of Bi_2_Te_3_ nanosheets using the solvothermal method relies on the appropriate addition of PVP. Due to the reaction of TeO_2_ with NaOH, producing TeO_3_^2−^, under pyrolysis conditions, Te elemental form is initially generated between TeO_3_^2−^. PVP K30, with a molecular weight (40,000) larger than Bi_2_Te_3_, acts as a spatial hindrance, suppressing the overly rapid combination of TeO_3_^2−^ ions and the formation of Te elemental form. When the amount of added PVP is inappropriate (0.8 g), it still leads to an excessive generation of Te. However, when 1 g of PVP is added, the solvothermal synthesis of Bi_2_Te_3_ nanosheets exhibited a complete hexagonal morphology, as shown in Figure 2a. The surface of Bi_2_Te_3_ nanosheets synthesized by the solvothermal method is smooth after being cleaned by acetone and ethanol. There was no adhesion between the hexagonal nanosheets. After particle size analysis, the edge length of the hexagonal Bi_2_Te_3_ nanosheets was determined to be 383 nm, and the thickness of the sheet was 20 nm (Figure 2b). Bi_2_Te_3_ has a rhombohedral layered structure, where atomic layers are arranged along the C-axis in a Te-Bi-Te manner. Van der Waals forces bind the adjacent Te layers, while intra-layer atoms are bonded by atomic bonds.

The chemical forces between atomic bonds are much stronger than van der Waals forces. Therefore, Bi_2_Te_3_ tends to grow along the layers with stronger bonding forces, resulting in the natural growth of Bi_2_Te_3_ in a layered sheet-like structure. The crystal unit cell structure of Bi_2_Te_3_ nanosheets is depicted in Figure 3. Each rectangular box represents a unit cell, and each nanosheet is composed of multiple unit cells.

Dispersing Bi_2_Te_3_ nanosheets in the aqueous solvent of PEDOT:PSS. Composite film was prepared by casting and exhibited a smooth surface. From Figure 2c, it is observed that Bi_2_Te_3_ nanosheets are embedded parallel to the *Z*-axis direction within the PEDOT film. The continuous phase formed by the crystallization of PEDOT presents a layered structure that stacks with the Bi_2_Te_3_ nanosheets (Figure 2d). The film thickness is approximately 10 μm. The layers are closely packed, promoting the efficient transport of charge carriers. When materials with a high Seebeck coefficient are introduced, the Seebeck coefficient of the film also increases accordingly. Simultaneously, the nanosheet size effect in the film can enhance phonon scattering. Phonon scattering is the process of energy exchange between carriers and phonons (energy quanta of the lattice vibrational law) in crystals, which is temperature and carrier transport dependent. Phonon scattering effectively diminishes the thermal conductivity of the material, resulting in an enhancement of the thermoelectric performance of the flexible film.

Transmission electron microscopy (TEM) was employed to examine the morphology and crystal structure of Bi_2_Te_3_ nanosheets. As illustrated in Figure 4a,b, TEM images of the Bi_2_Te_3_ nanosheet powder unveil the synthesis of well-defined hexagonal-shaped nanosheets featuring a side length of approximately 383 nm. Although the size of Bi_2_Te_3_ nanosheets exhibits some non-uniformity and lacks a distinct preferred orientation, the atomic structure remains evident. The electron diffraction pattern (Figure 4c) attests to the single-crystalline nature of the synthesized nanosheets. In Figure 4d, EDS analysis reveals a uniform distribution of Te and Bi elements, further confirming the sheet-like material as Bi_2_Te_3_ nanosheets. The introduction of PVP in this method plays a pivotal role. In this reaction, Te undergoes preferential reduction in ethylene glycol. The high electronegativity of O and N in polyvinylpyrrolidone imparts strong coordination capabilities. Through electrostatic forces, O and N interact with Te atoms, therefore impeding the overly rapid combination of TeO_3_^2−^. This inhibition enhances the likelihood of subsequent reactions, particularly the combination of Te and Bi atoms. Due to the substantial binding force between Te and Bi atoms, the growth rate within the crystal layers surpasses that along the c-axis. This phenomenon leads to the formation of well-shaped and smooth-surfaced nanosheets.

### 3.2. Phase Structure Analysis of PEDOT@Bi_2_Te_3_ Composite Films

The crystal structure and analysis of the solvothermal synthesized Bi_2_Te_3_ nanosheets and PEDOT@Bi_2_Te_3_ composite films were investigated using XRD. The test results are exhibited in Figure 5, where the diffraction peak positions and intensities of the synthesized powder were compared to PDF (#15-0863) for trigonal crystal system Bi_2_Te_3_, confirming that the synthesized powder consists of trigonal crystal Bi_2_Te_3_ nanosheets. The electron diffraction patterns are instrumental in determining the crystal structure. The diffraction pattern in Figure 4c was analyzed using Digital Micrograph to obtain the interplanar spacings (±0.01 nm) of the nanosheets, which are 0.320 nm, 0.237 nm, 0.219 nm, 0.203 nm, 0.181 nm, 0.161 nm, 0.149 nm, and 0.140 nm, corresponding to the crystal planes (015), (1010), (110), (0015), (205), (0210), and (1115) of Bi_2_Te_3_. This aligns with the crystal planes corresponding to the diffraction peaks at 2θ of 27.7°, 37.8°, 41.1°, 44.6°, 50.1°, 57.1°, 62.3°, and 66.9° in XRD, confirming the solvothermal synthesized black powder consists of trigonal crystal Bi_2_Te_3_ nanosheets. The diffraction peaks of the PEDOT@Bi_2_Te_3_ film did not exhibit significant changes in position compared to the diffraction peaks of the Bi_2_Te_3_ powder. While the crystal diffraction peaks are intense, a noticeable bread-loaf-shaped peak appears, attributed to PEDOT as an organic polymer [28].

Raman spectroscopy utilizes laser-induced non-elastic scattering to obtain a spectrum of different frequencies of scattered light from the chemical bonds through stretching and bending vibrations, therefore determining the molecular structure and crystal of composite materials. Figure 6a displays the Raman spectra of a series of PEDOT@Bi_2_Te_3_ films. After DMSO treatment, the original PEDOT exhibited a distinct characteristic peak at 1442 cm^−1^, indicating effective doping with DMSO and detachment of PSS anions. The feature peak at 1442 cm^−1^ represents the symmetric stretching of C_α_=C_β_ in the combined molecular chains of PEDOT. After Bi_2_Te_3_ doping, the vibrational peaks in the series of PEDOT@Bi_2_Te_3_ samples become more pronounced, especially evident with 5 wt.% and 10 wt.% Bi_2_Te_3_ nanosheet doping. The peak shifts to 1437 cm^−1^ in the PEDOT@Bi_2_Te_3_ films, and the half-width of the main peak narrows, indicating a partial reaction between PEDOT and Bi_2_Te_3_, rendering PEDOT more neutral [29]. The A_1_^2g^ of Bi_2_Te_3_, relative to C_α_=C_β_ of PEDOT, exhibits a weaker intensity in the main peak, but the presence of the interlayer vibration peak at 127 cm^−1^ is noticeable [15]. Changes in the FWHM of the main peak demonstrate the existence of charge transfer between Bi_2_Te_3_ and PEDOT, signifying a chemical bond between the inorganic nanosheets and the organic polymer. Figure 6b presents the FT-IR spectra of the composite films. Compared to the vibrational peaks of PEDOT, all composite materials exhibit characteristic peaks of PEDOT, albeit relatively weaker. This is attributed to the distribution of inorganic fillers within the polymer. The infrared vibration characteristic peaks of the composite films are consistent with previous literature reports [30]. The vibration absorption peaks observed at 1529 cm^−1^ and 1371 cm^−1^ are attributed to the stretching vibrations of C-C in PEDOT and the stretching vibrations of C=C in thiophene, respectively. The vibration absorption at 1350 cm^−1^ is presented only in the spectrum of the composite film and is not found in the PEDOT spectrum. According to the literature by Jiang et al., the absorption peak observed at 1385 cm^−1^ aligned with the absorption peak characteristic of Bi_2_Te_3_ nanoparticles. Consequently, the vibration absorption at 1350 cm^−1^ is likely associated with the absorption peak of the composite material [30]. Additionally, the absorption peak at 1263 cm^−1^ corresponds to the stretching vibration of S-O in PSS, and the absorption peak at approximately 1010 cm^−1^ corresponds to the stretching vibration of C-S bonds in the thiophene ring of PEDOT. The characteristic peaks at 860 cm^−1^ and 584 cm^−1^ correspond to the deformation vibrations of ethoxy groups in the polymer molecules. The infrared absorption spectrum suggests the possible existence of chemical bond composites between PEDOT and Bi_2_Te_3_ in the composite film, but further analysis of their interface binding modes is required through XPS.

### 3.3. Interface Structure Analysis of PEDOT@Bi_2_Te_3_ Composite Films

To further understand the elemental states and bonding modes at the interface of the organic-inorganic composite material, X-ray photoelectron spectroscopy (XPS) was employed for the characterization analysis of the PB-10. The XPS spectra of the PB-10 films are shown in Figure 7. The bonding states and elemental contents between Bi 4f, Te 3d, S 2p, and C 1s in the composite films were detected efficiently, but the characteristic peaks were relatively weak because the penetration depth of XPS into the films is less than 10 nm. Bi_2_Te_3_ nanosheets were filled with the PEDOT molecular. Figure 7a presents the Gaussian fitting spectra of Bi 4f, but the binding energy of the PSS compound is at 168 eV and 169.1 eV, indicating the presence of the PSS compound within the characteristic spectrum. Compared to the fitted spectra of Bi 4f in Bi_2_Te_3_ powders in Figure 7e, only the Bi 4f_5/2_ energy level of Bi-Te located at 163.7 eV corresponding to the PSS compounds can be observed, the low filler amount and the low Bi 4f_5/2_ intensity. The characteristic peaks located at 165.1 eV and 159.9 eV correspond to the Bi 4f_5/2_ and Bi 4f_7/2_ energy levels of Bi-S, respectively, due to the reaction between the SO_3^−^_ of the predominant PSS polymer and Bi_2_Te_3_ at the interface [24]. In addition, Bi_2_Te_3_ powders stored in sealed bags are prone to oxidation, which is evident in the oxidation peaks at 164.4 eV and 159.1 eV. The oxidation peaks disappeared after complexation with PEDOT. In the Gaussian fitting spectra of Te 3d (Figure 7b), the characteristic peaks at 587.1 eV and 576.8 eV represent the binding energies of Te 3d_3/2_ and Te 3d_5/2_, matching the orbital levels of Bi-Te in the Bi 4f spectrum. Figure 7c revealed the fitted profiles of S 2p. The characteristic peaks with binding energies of 169.1 eV and 167.8 eV correspond to 2p_1/2_ and 2p_3/2_ for PSS, 165.1 eV and 163.8 eV for PEDOT, and the characteristic peak appearing at binding energy 159.9 eV corresponds to the interface-generated S-Bi. The PEDOT film of S 2p was not found to have a characteristic peak (Figure 7i), indicating that the Bi_2_Te_3_ nanosheets in the composite film effectively bond with the PEDOT molecules to produce chemical bonds, and the reaction occurring at the interface may trigger the defective energy level, which has an impact on the charge transfer efficiency. Figure 7d indicates the high-resolution fitted profiles of C 1s, and the fitted peaks with binding energies located at 284.8 eV, 286.3 eV, and 287.9 eV correspond to C-C, C-S, and conjugated π-π in PEDOT, respectively. Combined with the C 1s spectra of PEDOT (Figure 7g), the embedding of Bi_2_Te_3_ nanosheets produces changes in the film structure with an increase in the C-S bond and conjugated π-π content, which revealed that the PSS is separated. The charge transfer efficiency of the composite film will be enhanced.

### 3.4. Thermoelectric Property and Flexibility of PEDOT@Bi_2_Te_3_ Composite Films

The authors should discuss the results and how they can be interpreted from the perspective of previous studies and the working hypotheses. The findings and their implications should be discussed in the broadest context possible. Future research directions may also be highlighted. The free-standing films of PEDOT@Bi_2_Te_3_ prepared by the tape-casting method were subjected to electrical property and Seebeck coefficient measurements using a thermoelectric performance testing system. The conductivity and Seebeck coefficient were determined as the mean of three measurements taken from different samples. As shown in Figure 8, the study systematically investigated the influence of the doping ratio of Bi_2_Te_3_ nanosheets on the thermoelectric performance of flexible films. It is observed that the conductivity of the free-standing composite film prepared by casting is higher than that of the film with only DMSO added (Figure 8a). The conductivity reaches its maximum value of 684 S/cm^−1^ when the mass fraction of Bi_2_Te_3_ nanosheets is 5 wt.%. As the mass fraction of Bi_2_Te_3_ nanosheets increases, the conductivity of the composite film gradually decreases, in accordance with the results presented by Wang et al. [26]. The addition of Bi_2_Te_3_ elevated the carrier concentration to five times the original (Figure 8d and the conductivity of the films increased abruptly, which was related to the narrowing of the forbidden bandwidth of the composites. While the gradual decrease in conductivity can be attributed to the increase of inorganic fillers with weak electrical conductivity, the carriers on the PEDOT molecular chains are hindered from transporting between the polymer chains, making the carrier transport more difficult, and the charge transfer efficiency is reduced making the conductivity of the PB-25 film severely reduced and lower compared to the PEDOT film.

In addition, the flexibility of the composite films was investigated by bending experiments and the change in conductivity of the films prepared by doping 10 wt.% Bi_2_Te_3_ nanosheets in PEDOT matrix and PEDOT films without added inorganic fillers were measured by bending the surface of the test tube with a diameter of 10 mm, as shown in Figure 9. The composite film exhibited a 14% decrease in conductivity after 1500 bending cycles. The PEDOT film consisting of molecular chains is more flexible and has a stable conductivity compared to the composite film, which is caused by the fact that the nanosheets are brittle materials compared to the molecular chains. The films subjected to stress produced microcracks, causing a slight decrease in conductivity. The composite films were preserved in air without encapsulation, and the electrical properties were stable after many tests. The films were more flexible and durable.

The effective and reasonable filling of inorganic fillers in PEDOT is aimed to enhance the Seebeck coefficient of the film. As depicted in Figure 8b, with the increase in the mass ratio of Bi_2_Te_3_ nanosheets, the Seebeck coefficient of the film increases by 19%. The PEDOT film exhibited a Seebeck coefficient of only 15.8 μV/K, while at a mass fraction of 10 wt.% of Bi_2_Te_3_ nanosheets, the Seebeck coefficient reached 18.7 μV/K at 300 K. The Seebeck coefficient exhibits an inverse proportionality to the carrier concentration, with the highest carrier concentration corresponding to the lowest Seebeck coefficient. As the Bi_2_Te_3_ filler increases, PEDOT carrier transport is hindered, leading to carrier concentration being decreased. Subsequently, there was a decrease in conductivity. The Seebeck coefficient initially increases and then decreases. The construction of two-dimensional binary composite films effectively enhanced the thermoelectric properties, and the Seebeck coefficient and conductivity were positively correlated with the power factor of the composite films. Carrier concentration is positively correlated with conductivity and negatively correlated with the Seebeck coefficient, so the relationship between carrier concentration and conductivity and the Seebeck coefficient is illustrated in Table 1:

The Seebeck coefficient of the composite material is notably higher compared to that of the PEDOT film. This can be attributed not only to the simple mixing effect but also to the critical role of interface transport in influencing the ultimate thermoelectric properties of hybrid composite materials. The introduction of sheet-like Bi_2_Te_3_ between PEDOT molecules increased the interface, facilitating charge transfer at the S-Bi bond interface, leading to an elevation in carrier concentration and a reduction in the bandgap. This narrowing is beneficial for carrier transition. However, the introduction of Bi_2_Te_3_ can also introduce defects. Defects form energy barriers at the interface, affecting the relaxation time of hole carriers. The relaxation time of carriers varies as a function of energy, forming a cutoff energy in carrier transport. When the energy of hole carriers is slightly lower than the cutoff energy, these carriers cannot participate in the charge transport in the composite material. Since low-energy hole carriers weaken the Seebeck coefficient, the scattering generated by the interface composite will prevent low-energy hole carriers from participating [31]. The association between the Seebeck coefficient and the cutoff energy can be expressed using Equation (1):(1)$$Se=A[G\left(E_{c}\right)-E_{F}]$$
where *A* is a constant, $E_{c}$ is the cutoff energy, and $G\left(E_{c}\right)$ remains greater than $E_{c}$ and $E_{F}$ and is a monotonically increasing function with the growth of $E_{c}$. Thus, as the energy level $E_{c}$ grows, the Seebeck coefficient of the composite material experiences an increase [32,33].

The dependence of relaxation time on energy increases the uncertainty of hole carrier transport at the Fermi level, resulting in an augmentation of the Seebeck coefficient. Owning to the interface energy barrier between PEDOT and Bi_2_Te_3_, the nanosheets in the hybrid material exhibit a larger interface volume ratio, providing more positions for a selective scattering of low-energy carriers, leading the carrier concentration decreased and the Seebeck coefficient increase. The continuous increase in the amount of Bi_2_Te_3_ did not further enhance the Seebeck coefficient, possibly because the interface composite between PEDOT and monodispersed Bi_2_Te_3_ nanosheets was well-formed, disrupting the connectivity between Bi_2_Te_3_ particles. Therefore, the rise in the Seebeck coefficient was not particularly noticeable. The power factor of the free-standing PEDOT@Bi_2_Te_3_ hybrid film with varying Bi_2_Te_3_ content is demonstrated in Figure 8c. The power factor of the original sample was 11.9 μW/mK^2^ at 300 K. When the Bi_2_Te_3_ content was 5 wt.%, the power factor of PB-5 was 20 μW/mK^2^. The power factor begins to gradually increase. When the Bi_2_Te_3_ content increased to 10 wt.%, PB-10 had the highest power factor of 22.3 μW/mK^2^. However, the Bi_2_Te_3_ doping exceeded 10 wt.%, the power factor showed a decreasing trend. With the continuous increase in doping amount, the power factor decreases to 15.3 μW/mK^2^. This phenomenon is mainly attributed to changes in conductivity. The interface binding of composite materials narrows the bandgap, making charge transfer easier and resulting in an elevation of hole carrier concentration and a boost in conductivity. The Seebeck coefficient decreased, but it was relatively large compared to the original film. This indicated that an appropriate organic-inorganic hybridization method improved the thermoelectric performance of free-standing flexible films.

### 3.5. Temperature-Dependent of Thermoelectric Properties

The Seebeck coefficient and conductivity serve as pivotal parameters influencing the thermoelectric performance. Conductivity is directly proportional to carrier concentration and inversely proportional to the Seebeck coefficient. Therefore, examining the temperature-dependent behavior of thermoelectric parameters and the thermal motion of carriers, concentration, and mobility at various temperatures can unveil the underlying mechanism responsible for the enhanced thermoelectric performance observed in the free-standing PEDOT@Bi_2_Te_3_ composite films achieved through organic/inorganic hybridization. Figure 10 illustrates the trends observed in the conductivity, Seebeck coefficient, and power factor of PEDOT@Bi_2_Te_3_ composite films across the temperature range of 30 to 110 °C, depicting varying proportions of Bi_2_Te_3_ nanosheet doping. As the temperature increased, the original film showed a relatively small change in electrical conductivity. PB-5 exhibited the highest electrical conductivity, gradually increasing with temperature, while PB-10 shows a slightly lower electrical conductivity with a decreasing trend at higher temperatures, and PB-25 also decreases, as illustrated in Figure 10a. This may be related to the excessive doping of Bi_2_Te_3_ nanosheets in PEDOT.

When composite films with varying mass ratios of Bi_2_Te_3_ to PEDOT are prepared at the same working temperature, the Seebeck coefficient remains relatively unchanged. In contrast, the Seebeck coefficient of the original film continues to increase. This indicated a change in the internal conduction mechanism of the film, suggesting temperature is no longer the primary factor affecting carrier concentration. Figure 10b indicates that the Seebeck coefficient of the composite film is solely associated with the doping amount of Bi_2_Te_3_ nanosheets. This is noteworthy considering the intrinsic conductivity of PEDOT:PSS, which falls within the range of 0.028 to 0.48 W/mK [34]. The addition of nanosheets increases the interface phonon scattering, reducing the thermal conductivity of the film to 0.07 W/mK [26]. This decrease could be attributed to the inadequate thermal conductivity of the conductive polymer and the hindrance caused by interface scattering, impeding the heat transport process. The different phonon densities and velocities between inorganic fillers and conductive polymers cause phonon mismatch, leading to the thermal conductivity is decreased. The in-situ interface reaction between PEDOT and Bi_2_Te_3_ does not affect the transport of hole carriers. Studies have indicated that the average free path of hole carriers in PEDOT is an order of magnitude shorter than that of phonons [34]. Therefore, the thermoelectric performance of the composite film is represented by the power factor. As exhibited in Figure 10c, within the low-temperature working range, the trend of the power factor changing is consistent with the trend of the Seebeck coefficient. Since the gradual decline in conductivity beyond 70 °C, PB-10 only decreased by 2%. The Seebeck coefficient tends to stabilize, and the power factor of PB-10 at room temperature is 22.3 μW/mK^2^, while the temperature-sensitive power factor of PB-5 is 23.4 μW/mK^2^ at 70 °C, with only a 1.1 μW/mK^2^ difference between them. Therefore, the power factor of flexible films tends to stabilize with temperature variations, rendering them suitable for self-supporting thermoelectric film applications in low-temperature thermoelectric conversion.

ZT is negatively correlated with thermal conductivity. According to the available literature [35], the intrinsic thermal conductivity of PEDOT is only 0.028–0.48 W/mK, which is much less than 1. The introduction of bismuth telluride nanosheets facilitates phonon scattering, which reduces the thermal conductivity of the lattice occupying 90% of the thermal conductivity, which, in combination with the low thermal conductivity of the polymers, results in an effective reduction of thermal conductivity of the composite film [26]. The ZT was calculated from the reported thermal conductivity of 0.07 W/mK of the composite film and compared with the current free-standing composite thermoelectric films, which was useful in determining the potential application of the PEDOT@Bi_2_Te_3_ composite film in the field of thermoelectric devices. As is exhibited in Figure 10d, the ZT value of the composite film prepared by organic-inorganic hybridization is higher than that of the layered-coated composite films. Zheng et al. [8] introduced Bi_2_Te_3_ nanowires, enhancing both conductivity and the Seebeck coefficient is achieved. However, films prepared using the spin-coating method lack independent free-standing properties. In this study, the performance of PB-10 prepared by the flow casting of Bi_2_Te_3_ nanosheets into PEDOT is second only to it. Further exploration of the process and performance improvement mechanisms is needed to enhance the thermoelectric performance of composite films.

### 3.6. Mechanism of Enhanced Thermoelectric Performance in PEDOT@Bi_2_Te_3_ Films

Carrier concentration and mobility are crucial parameters reflecting the electrical conductivity of semiconductor materials. The dependence on temperature is illustrated in Figure 11. The carrier concentration is used to determine the conductivity of semiconductors in the charge transport mechanism of semiconductors and is determined by Equation (2). The carrier concentration is positively correlated with the conductivity and negatively correlated with the Seebeck coefficient, which is also positively correlated with the defect energy level, as shown in Equation (1), and the charge transfer mechanism of the composite film is analyzed according to the structure of the composite material.
(2)σ=nqμ
(3)$$S=\frac{{8\pi }^{2}k_{B}^{2}}{4R}m^{*}T{\left(\frac{\pi }{3n}\right)}^{2/3}$$
where σ is the conductivity, n is the carrier concentration of PEDOT@Bi_2_Te_3_, q is the electron charge, $k_{B}$ is 1.38 × 10^−23^ J/K, and $m^{*}$ is the effective mass of charge. The effective mass of the charge is affected by doping, and the conductivity and the Seebeck coefficient are changed.

According to classical charge transport theory, the mobility of carriers is inversely proportional to temperature, showing a linear trend, decreasing as T increases in a T^−3/2^ fashion. The relationship between carrier mobility and concentration in the original film with temperature is shown in Figure 11a, consistent with transport theory. PEDOT@Bi_2_Te_3_ composite film prepared by organic-inorganic hybridization, hole mobility decreases and then increases with temperature. PB-5 and PB-10 exhibit nonlinear changes, while PB-25 has a lower carrier concentration and weaker conductivity but a continuously increasing mobility trend. This is because an excessive introduction of weakly conductive Bi_2_Te_3_ nanosheets reduces the effective charge mass inside the film due to carrier scattering, resulting in enhanced carrier mobility and reduced concentration, consequently leading to decreased conductivity in PB-25. However, an appropriate amount of Bi_2_Te_3_ nanosheet doping reduces the energy barrier, resulting in an elevated carrier concentration [11]. When the working temperature rises to 360 K, mobility decreases, low-energy carriers undergo phonon scattering, and the charge mass decreases, while high-energy carriers persist in transporting within the composite film, causing an increase in carrier concentration. This phenomenon is ascribed to the interface energy-filtering effect in the composite films.

The PEDOT@Bi_2_Te_3_ composite films prepared by casting make the Bi_2_Te_3_ nanosheets orientated. The PEDOT molecules and Bi_2_Te_3_ nanosheets are arranged in an orderly and regular manner, and the composite interfaces are bound by Bi-S bonds. The forbidden bandwidth shrinks from 1.7 eV to 1.63 eV at 10 wt.% of Bi_2_Te_3_ in the composite film, as shown in Figure 12. The Hall-effect test results also showed that making Bi_2_Te_3_ as an inorganic filler composite with PEDOT increased the hole carrier concentration of the film (Figure 11b). The introduction of the topological insulator Bi_2_Te_3_ produces defect energy levels, which are positively correlated with the Seebeck coefficient, as shown in Equation (1). When PEDOT transports charge to Bi_2_Te_3_, it cannot participate in carrier transport due to defect energy level trapping, and when the temperature increases to 360 K, the charge trapped by the defect energy level is sufficient to break away from the confinement and jump to the conduction band, resulting in a decrease in carrier concentration and an increase in mobility.

The composite films of PEDOT@Bi_2_Te_3_ prepared by casting enabled the Bi_2_Te_3_ nanosheets to be orientated, the PEDOT molecules and Bi_2_Te_3_ nanosheets to be arranged in an orderly and regular manner, and the composite interfaces to be bound by Bi-S bonds. The forbidden bandwidth was reduced from 1.7 eV to 1.63 eV at 10 wt.% Bi_2_Te_3_ in the composite film, as shown in Figure 13. Hall-effect test results also indicated an increase in the hole carrier concentration of the films after making Bi_2_Te_3_ as an inorganic filler composite with PEDOT (Figure 11b). Introducing the topological insulator Bi_2_Te_3_ produces defect energy levels, and defect energy levels are positively correlated with the Seebeck coefficient, as shown in Equation (1). When PEDOT transported charge to Bi_2_Te_3_, it could not participate in carrier transport due to defect energy level trapping, and as the temperature increased to 360 K, the charge captured by the defect energy level was sufficient to break away from the confinement and jump to the conduction band, resulting in a decrease in carrier concentration and an increase in mobility.

## 4. Conclusions

In summary, high-performance PEDOT@Bi_2_Te_3_ composite thermoelectric films were successfully produced by a continuous casting process. Bi_2_Te_3_ low-dimensional nanosheets were synthesized by the solvothermal method using the spatial site resistance effect of polyvinylpyrrolidone. It was uniformly distributed in PEDOT by electrostatic self-assembly bonding. With a mass fraction of 10 wt.% of Bi_2_Te_3_, the composite film had a conductivity of 684 S/cm and a Seebeck coefficient of 18.7 μV/K, producing a power factor of 22.3 μW/mK^2^, which was two times higher than that of the pure-phase PEDOT film. For the composite films, the good thermoelectric properties may be attributed to the unique bismuth telluride nanosheets with quinone-based PEDOT leveling arrangement achieving high charge transfer efficiency, as well as the beneficial effect of bonding at the interface of PEDOT and Bi_2_Te_3_. To investigate the specific charge transfer mechanism in the binary composites, carrier concentration, and mobility were tested using a Hall-effect instrument to analyze the conductivity and Seebeck coefficient. Bi_2_Te_3_ doping led to an increase in the carrier concentration and mobility of the composite film, and enhanced conductivity. Meanwhile, defect energy levels were created in the composites, and the Seebeck coefficient increased. The conductivity of the composite film is decoupled from the Seebeck coefficient, and the power factor is improved by 68.9%. Furthermore, the film has good flexibility, and the conductivity decreases by only 14% after 1500 bending cycles. Such flexible thermoelectric films with free-standing properties have potential applications in the field of self-powered sensing using human thermal energy.

## Figures and Tables

**Figure 1 polymers-16-01979-f001:**
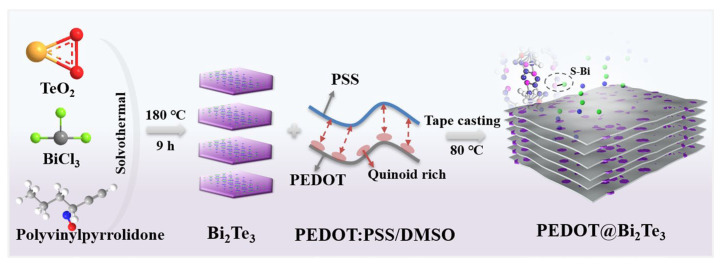
Schematic structure of PEDOT@Bi_2_Te_3_ composited films.

**Figure 2 polymers-16-01979-f002:**
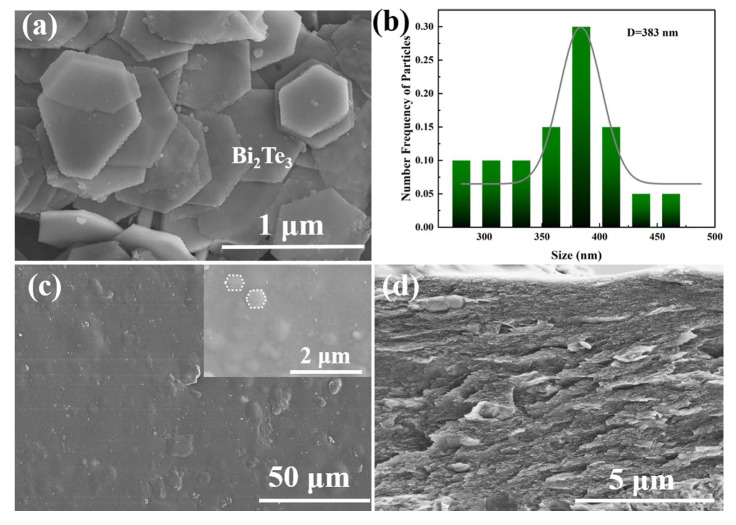
SEM images of Bi_2_Te_3_ nanosheets with 1g PVP (**a**); Particle size distribution statistics of Bi_2_Te_3_ nanosheets (**b**); Surface and cross-section of PEDOT@Bi_2_Te_3_ composited film (**c**,**d**).

**Figure 3 polymers-16-01979-f003:**
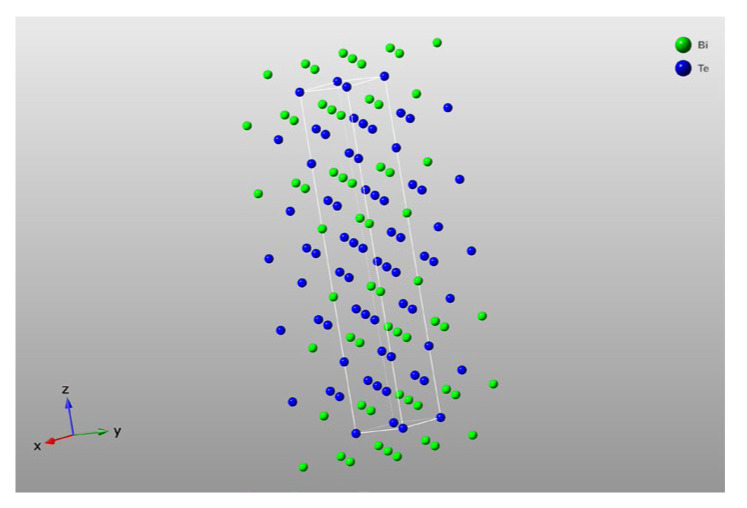
Crystal structure of Bi_2_Te_3_ nanosheets fabricated by solvothermal method.

**Figure 4 polymers-16-01979-f004:**
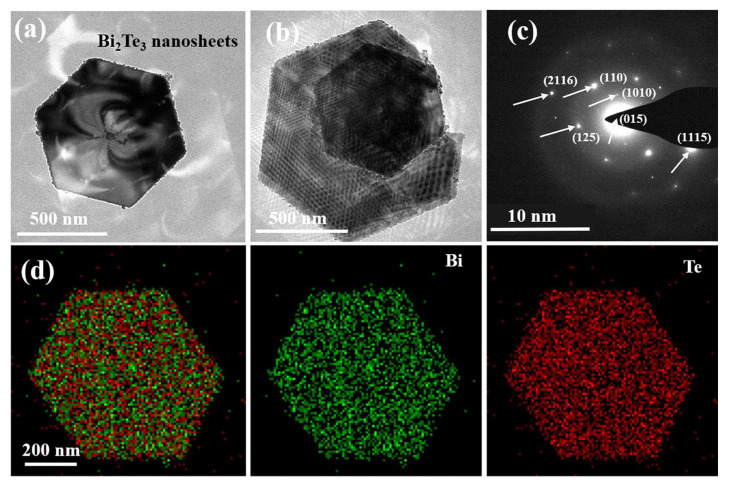
TEM images of Bi_2_Te_3_ nanosheets (**a**,**b**); SAED (**c**); EDS (**d**).

**Figure 5 polymers-16-01979-f005:**
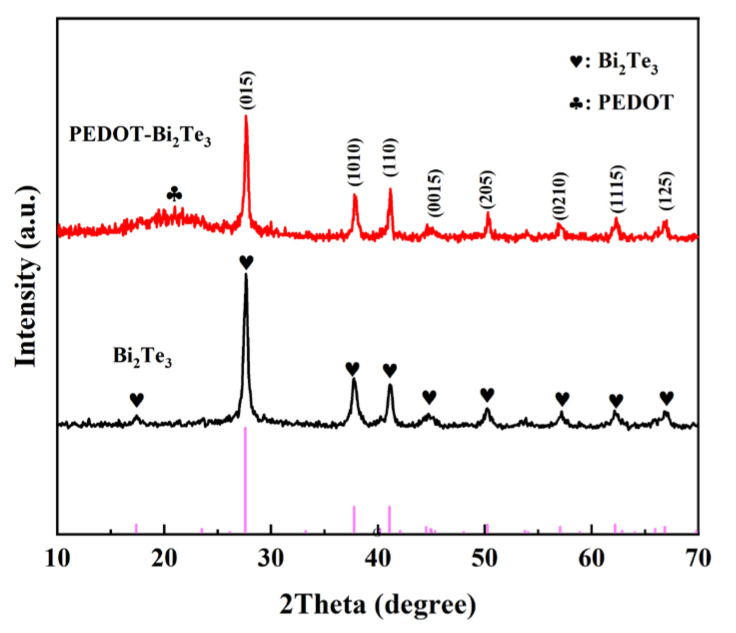
XRD spectrum of Bi_2_Te_3_ nanosheets and PEDOT@Bi_2_Te_3_ composite films.

**Figure 6 polymers-16-01979-f006:**
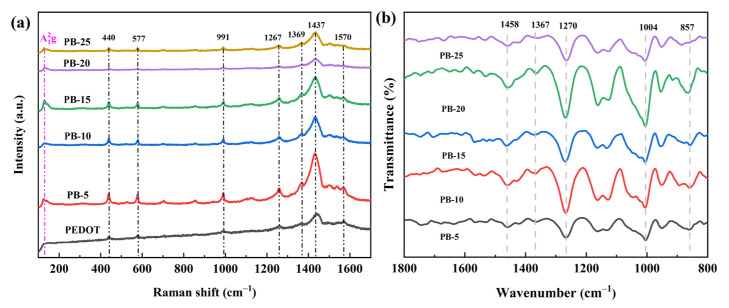
Raman spectra (**a**) and FTIR spectra (**b**) of a series of PEDOT@Bi_2_Te_3_ films.

**Figure 7 polymers-16-01979-f007:**
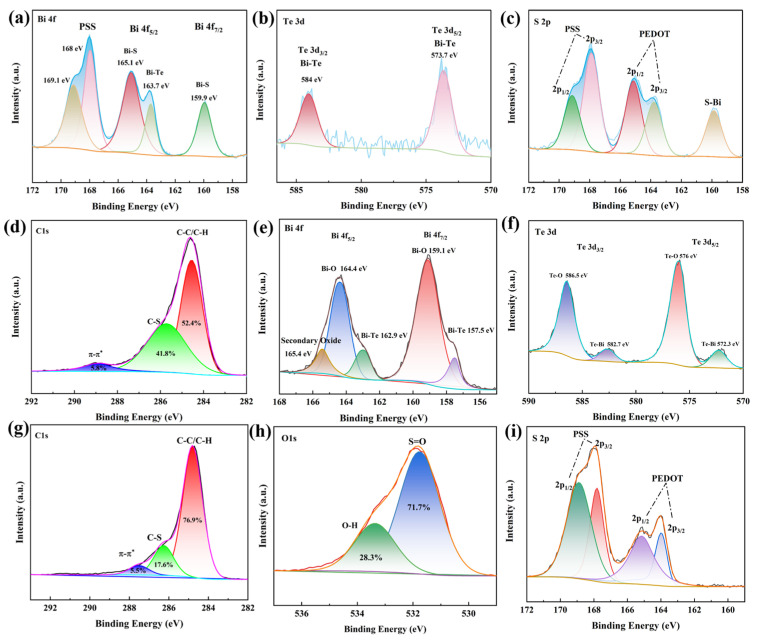
(**a**–**d**) Gaussian fitting spectra of Bi 4f, Te 3d, S 2p and C 1s for PB-10 (**e**,**f**) Bi 4f and Te 3d of Bi_2_Te_3_; (**g**–**i**) C 1s, O1s, and S 2p of PEDOT.

**Figure 8 polymers-16-01979-f008:**
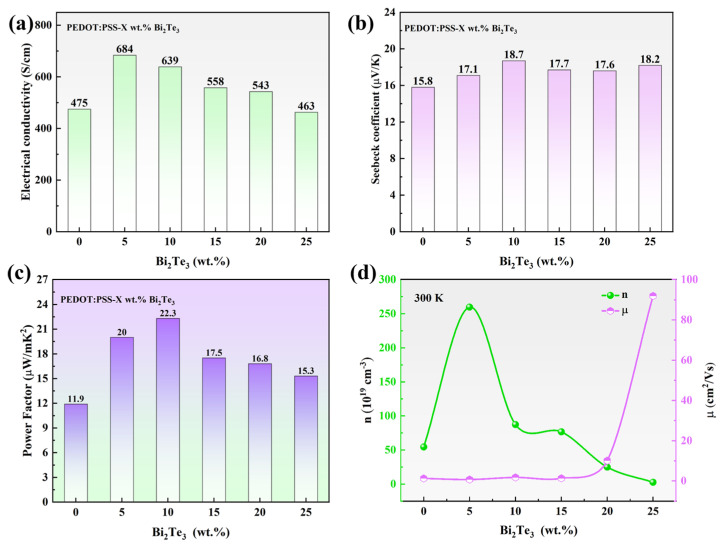
Thermoelectric properties of PEDOT nanofilms with different Bi_2_Te_3_ doping amounts: conductivity (**a**); Seebeck coefficient (**b**); Power factor of PSS nanocomposite films (**c**); Hall effect testing the mobility and carrier concentration of the composite film (**d**).

**Figure 9 polymers-16-01979-f009:**
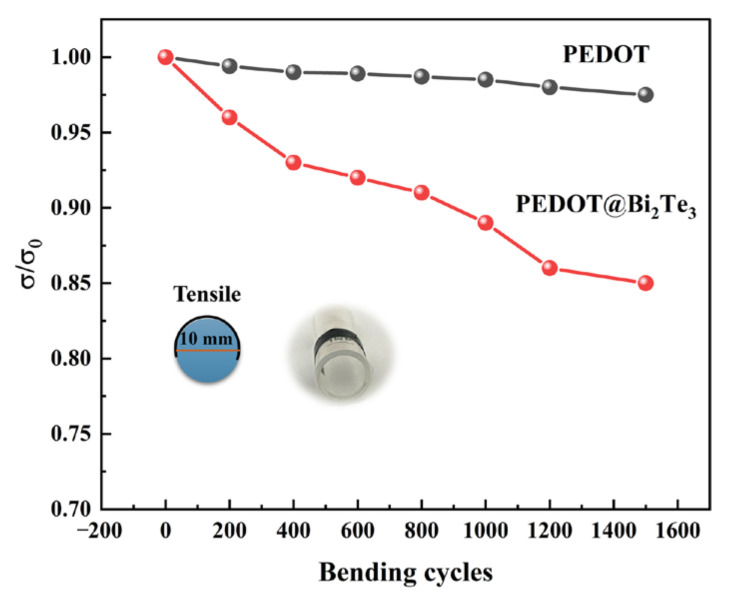
The ratio σ/σ0 as a function of the film bending cycle with a bending radius of 10 mm, and the upper block is the tensile bending.

**Figure 10 polymers-16-01979-f010:**
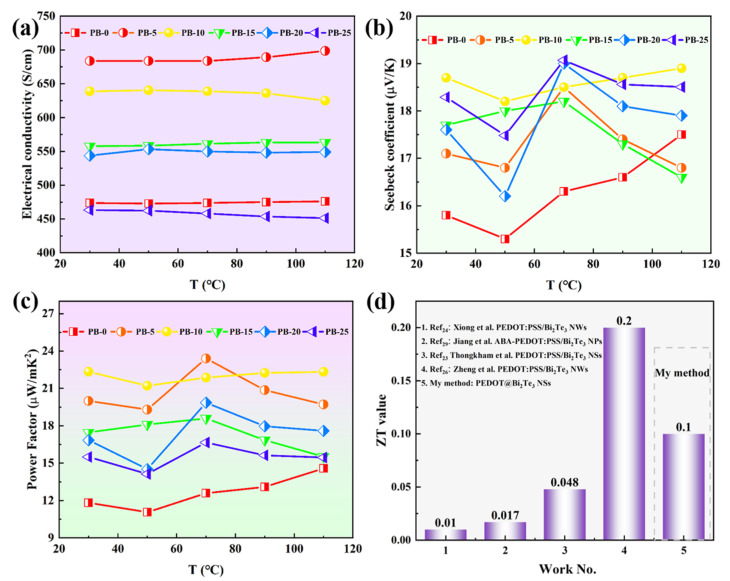
Temperature dependence of PEDOT@Bi_2_Te_3_ nanofilms: conductivity (**a**); Seebeck coefficient (**b**); Power factor (**c**); Comparison of ZT values for the free-standing composited films (**d**).

**Figure 11 polymers-16-01979-f011:**
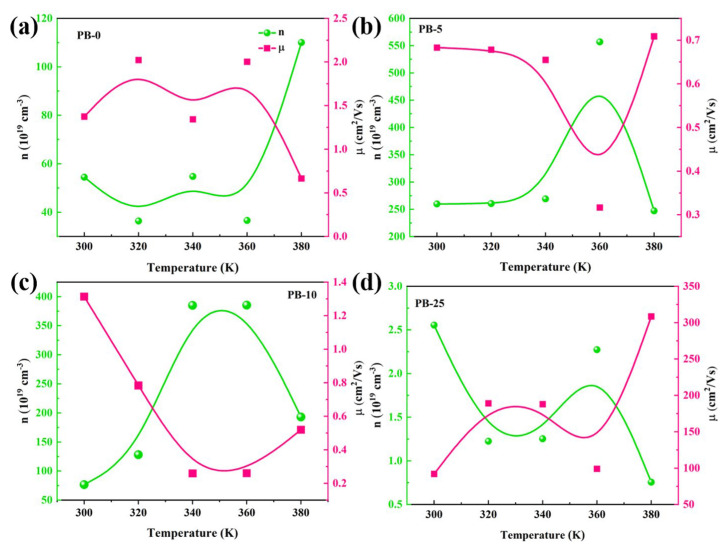
Hall effect testing results of the composite films with operating temperature. (**a**–**d**): PB-0, PB-5; PB-10; PB-25.

**Figure 12 polymers-16-01979-f012:**
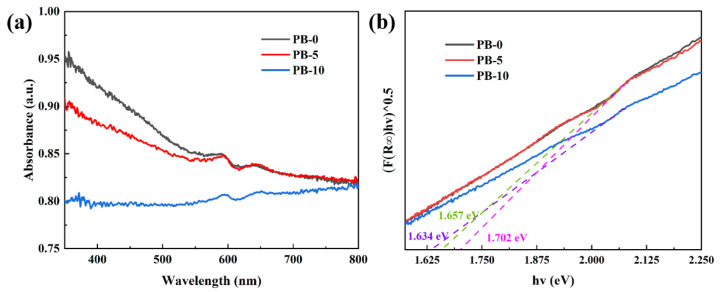
UV–Vis diffuse reflectance spectra (**a**) and corresponding bandgap energy (**b**) of PB-0, PB-5, and PB-10.

**Figure 13 polymers-16-01979-f013:**
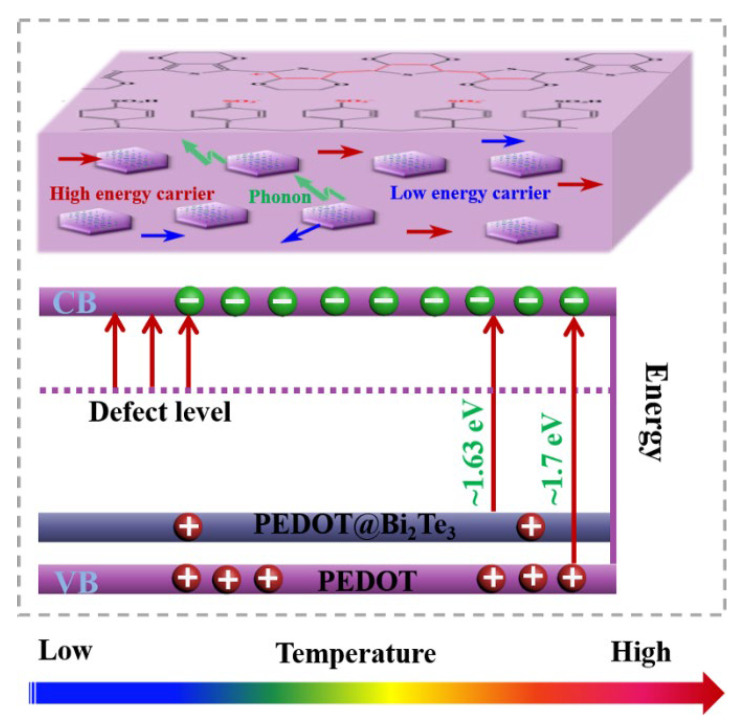
Schematic diagram of the thermoelectric improvement mechanism of PEDOT@Bi_2_Te_3_.

**Table 1 polymers-16-01979-t001:** Comparison of thermoelectric parameters for a series of composite films.

Sample	σ (S/cm)	S (μV/K)	PF (μW/mK^2^)	*n* (×10^19^ cm^-3^)	μ (cm^2^/Vs)
PB-0	475	15.8	11.9	54.4	1.3
PB-0	684	17.1	20	259.7	0.7
PB-0	639	18.7	22.3	87.3	1.8
PB-0	558	17.7	17.5	76.7	1.3

## Data Availability

The original contributions presented in the study are included in the article. Further inquiries can be directed to the corresponding author.

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
