# Peer review of "One Stone, Three Birds: Feasible Tuning of Barrier Heights Induced by Hybridized Interface in Free-Standing PEDOT@Bi2Te3 Thermoelectric Films"

_polymers, 2024, doi:10.3390/polym16141979_

Round 1
Reviewer 1 Report
Comments and Suggestions for Authors
The manuscript describes a process of preparation of PEDOT@Bi2Te3 composite films for thermoelectric application. The samples obtained were characterized by SEM, Raman- and Fourier-transform infrared spectroscopy, X-ray photoelectron spectroscopy. Conductivity, Seebeck coefficient, and power factor (PF), were evaluated, and additionally, the flexibility and sensitivity of the composite films were measured. As a result free-standing flexible films of PEDOT@Bi2Te3 were prepared; the thermoelectric performance of these films is modest compared to the literature (see Introduction).
The manuscript is well structured but results description, data discussion and interpretation have shortcomings.
General comments:
The Introduction Section presents only brief list of Bi2Te3-PEDEOT:PSS-based thermoelectric materials without in-depth discussion of the issues to be improved. The question under study is not well-defined and does not seem to be original. It is highly desirable to present in Introduction working hypotheses that can be used to advance the current knowledge or improve performance of materials of interest. Furthermore, the last paragraph of Introduction (lines 77-90) is more like an Abstract than an Introduction. The aim of the paper is not outlined. Major revision suggested.
The first four lines of the Conclusion Section (lines 491-494) have no meaning and look like recommendation to write a good paper. It is necessary to bring the contents of the “Conclusions” paragraph into line with the purpose of the work outlined in the revised “Introduction”
Specific comments:
It seems to be a good idea to give explanation for DMSO, PVP, PEDOT and PEDOT:PSS abbreviations. Moreover it is confusing because PVP may stand for Poly-4-vinylphenol or Polyvinylpyrrolidone or something else.
“Different masses of monodisperse Bi2Te3 nanosheets (line 105)…”. The paper does not provide particle size distribution and a description of isolation of “monodisperse Bi2Te3 “nanosheets. Revision suggested.
Figure 1 does not give “…the detailed experimental procedure…” (line 110) of PEDOT@Bi2Te3 flexible films preparation.
“The successful synthesis of Bi2Te3 nanosheets using the solvothermal method relies 131 on the appropriate addition of PVP” (lines 131-132). It is difficult to judge about “appropriate” or “inappropriate” amount of PVP based on only two points - 0.8 and 1 g.
The discussion of the reasons for the appearance of Te (lines 131-137) has no experimental proofs. Appropriate reference needed.
Figures 3 and 4d are redundant and can be excluded. The assignment of needle-like crystals to Te is unsubstantiated. A Figure with distribution of Te and Bi (EDS) for figs. 2 a,c,d is highly desirable. Revision suggested.
“The film was soaked in water, peeled off from the glass substrate, and dried at 80 ° C for 4 h to obtain a smooth-surfaced flexible composite film with a thickness of 10 μm.” (lines 108-110). This contradicts Fig.2d (cross section) where the thickness obviously exceeds 100 μm. Revision suggested.
To calculate ZT it is necessary to know thermal conductivity coefficient k. It is not clear how k was measured or evaluated (lines 412-414). k value is desirable.
Author Response
Comments to the Author
- The Introduction Section presents only brief list of Bi2Te3-PEDEOT:PSS-based thermoelectric materials without in-depth discussion of the issues to be improved. The question under study is not well-defined and does not seem to be original. It is highly desirable to present in Introduction working hypotheses that can be used to advance the current knowledge or improve performance of materials of interest. Furthermore, the last paragraph of Introduction (lines 77-90) is more like an Abstract than an Introduction. The aim of the paper is not outlined. Major revision suggested.
- Response: Thank you very much for your valuable and constructive advice. Regarding this issue, we have modified and enriched the introduction in the manuscript as follows: In recent years, most of the composite films studied were prepared on glass substrates or silicon wafers using spin coating or vapor phase deposition processes. The thickness of the film is less than 100 nm, which is brittle and not suitable for mass production and preparation. Thermoelectric conversion performance needs to be improved. Facing the actual demand of wearable applications, the thermoelectric and mechanical properties of these composite films still need to be improved. Meanwhile, regarding the emerging composite systems in two-dimensional configurations, previous studies have mainly focused on film preparation techniques and post-processing property enhancement, and their specific features in terms of the structural morphology of materials and carrier transport behaviors have not yet been comprehensively analyzed. In this work, we successfully prepared Bi2Te3 nanosheets by solvothermal method, and used the spatial site resistance effect of Polyvinylpyrrolidone (PVP) to modulate the morphological design of Bi2Te3. Free-standing PEDOT@Bi2Te3 films were prepared by combining PEDOT with them through electrostatic self-assembly method and casting film-forming process. Composite films produced not only achieved a uniform distribution of Bi2Te3 in PEDOT, but also integrated the high conductivity and a large Seebeck coefficient of the two components, resulting in improved thermoelectric properties of the composite films. Charge transfer mechanisms considering composition and orientation are proposed for this specific 2D nanosheet binary composite system. Direct interfacial effects of the materials are also discussed. With high thermoelectric properties and excellent flexibility, the PEDOT@Bi2Te3 composite films developed have promising applications in wearable thermoelectric devices or sensor applications.
- The first four lines of the Conclusion Section (lines 491-494) have no meaning and look like recommendation to write a good paper. It is necessary to bring the contents of the “Conclusions” paragraph into line with the purpose of the work outlined in the revised “Introduction”.
- Response: Thank a lot for your kind reminding. We have deleted it and the conclusions were also modified based on the contents of the introduction. Regarding this issue, we have modified and enriched the conclusion in the manuscript as follows: In summary, high-performance PEDOT@Bi2Te3 composite thermoelectric films were successfully produced by a continuous casting process. Bi2Te3 low-dimensional nanosheets were synthesized by solvothermal method using the spatial site resistance effect of PVP. It was uniformly distributed in PEDOT by electrostatic self-assembly bonding. With a mass fraction of 10 wt.% of Bi2Te3, the composite film had a conductivity of 684 S/cm and a Seebeck coefficient of 18.7 μV/K, producing a power factor of 22.3 μW/mK2, which was two times higher than that of the pure-phase PEDOT film. For the composite films the good thermoelectric properties may be attributed to the unique bismuth telluride nanosheets with quinone-based PEDOT levelling arrangement achieving high charge transfer efficiency, as well as the beneficial effect of bonding at the interface of PEDOT and Bi2Te3. To investigate the specific charge transfer mechanism in the binary composites, carrier concentration and mobility were tested using Hall effect to analyze the conductivity and Seebeck coefficient. Bi2Te3 doping led to an increase in the carrier concentration and mobility of the composite film, and enhanced conductivity. Meanwhile, defect energy levels were created in the composites and the Seebeck coefficient increased. Conductivity of the composite film is decoupled from the Seebeck coefficient and the power factor is improved by 68.9%. Furthermore, the film has good flexibility and the conductivity decreases by only 14% after 1500 bending cycles. Such flexible thermoelectric films with free-standing properties have potential applications in the field of self-powered sensing using human thermal energy.
- It seems to be a good idea to give explanation for DMSO, PVP, PEDOT and PEDOT:PSS abbreviations. Moreover, it is confusing because PVP may stand for Poly-4-vinylphenol or Polyvinylpyrrolidone or something else.
- Response: Thank you very much for your affirmative attitude and valuable comments. PVP stands for Polyvinylpyrrolidone in the manuscript. We have marked Polyvinylpyrrolidone as PVP in the introduction, and any subsequent occurrences of PVP in the text represent Polyvinylpyrrolidone.
- “Different masses of monodisperse Bi2Te3 nanosheets (line 105)…”. The paper does not provide particle size distribution and a description of isolation of “monodisperse Bi2Te3 “nanosheets. Revision suggested.
- Response: Thank a lot for your kind reminding. It’s significant to offer the particle size distribution and a description of Bi2Te3 The surface of Bi2Te3 nanosheets synthesized by solvothermal method is smooth after being cleaned by acetone and ethanol. There was no adhesion between the hexagonal nanosheets as shown in Figure 2a. After particle size analysis, the edge length of the hexagonal Bi2Te3 nanosheets was determined to be 383 nm and the thickness of the sheet was 20 nm. Descriptions of the particle size distribution and surface morphology of Bi2Te3 nanosheets have been added in the main text.
-
Figure 2. SEM images of Bi2Te3 nanosheets with 1 g PVP (a); Particle size distribution statistics of Bi2Te3 nanosheets (b); Surface and cross-section of PEDOT@Bi2Te3 composited film (c, d)
- Figure 1 does not give “…the detailed experimental procedure…” (line 110) of PEDOT@Bi2Te3 flexible films preparation.
- Response: Thanks so much to you and please allow us to present our apologies for our carelessness. We checked carefully again and found that the Figure 1 is a schematic structure of the synthesized PEDOT@Bi2Te3 composite film. The detailed experimental preparation procedure was consistent with that described in the manuscript. We have modified it in the manuscript as follow: The schematic structure of PEDOT@Bi2Te3 composited film is depicted in Figure 1.
- “The successful synthesis of Bi2Te3 nanosheets using the solvothermal method relies 131 on the appropriate addition of PVP” (lines 131-132). It is difficult to judge about “appropriate” or “inappropriate” amount of PVP based on only two points - 0.8 and 1 g.
- Response: Thank you very much for your kind reminding. Bi2Te3 nanosheets synthesized by solvothermal method were without impurity phase when the dosage of PVP was 10 g. After cleaning with acetone and ethanol, the hexagonal nanosheets are monodisperse and have a smooth surface, which is suitable for compounding with organic materials to avoid agglomeration. Regarding the content in Figure 2 has been modified in the manuscript.
- The discussion of the reasons for the appearance of Te (lines 131-137) has no experimental proofs. Appropriate reference needed.
- Thank you for your affirmative attitude and valuable comm The discussion of the reasons for the appearance of Te is based on experimental discussions. During solvothermal synthesis of Bi2Te3 nanosheets, rod-shaped Te appeared when the amount of PVP was 0.8 g, while the product after centrifugal washing showed completely hexagonal nanosheets when 1 g of PVP was added to the precursor. Thus, the discussion on solvothermal synthesis of Bi2Te3 nanosheets is substantiated.
- Figure 3 and Figure 4d are redundant and can be excluded. The assignment of needle-like crystals to Te is unsubstantiated. A Figure with distribution of Te and Bi (EDS) for figs. 2 a, c, d is highly desirable. Revision suggested.
- Thank you very much for your valuable advice. Bi2Te3 nanosheets synthesized by the solvothermal method have a completely hexagonal morphology and a lamellar structure with a thickness of 20 nm, which is attributed to the rhombohedral hexahedral lamellar structure of Bi2Te3. Each atomic layer is arranged along the C-axis in a Te-Bi-Te manner, with van der Waals force bonding between two adjacent Te layers and covalent bonding of atoms within the layers. As shown in Figure 3, the chemical force of covalent bonding is much larger than the van der Waals force. Therefore, Bi2Te3 is more likely to grow along the layers with high bonding force, and the naturally grown Bi2Te3 tends to be in the form of lamellae. Figure 4d illustrated the hexagonal crystals contain only two elements, Bi and Te, indicating the successful synthesis of Bi2Te3 crystals by solvothermal method.
- “The film was soaked in water, peeled off from the glass substrate, and dried at 80 °C for 4 h to obtain a smooth-surfaced flexible composite film with a thickness of 10 μm.” (lines 108-110). This contradicts Fig. 2d (cross section) where the thickness obviously exceeds 100 μm. Revision suggested.
- Thanks so much to your kind reminding and very sorry for our mistake. We found that the scale in Figure 2d had been mark incorrectly and we revised it. And now the caption and the explanation in the main text of Figure 2d is consistent. Figure 2d has been changed in the manuscript and the right figure is shown as follows:
Figure 2. SEM images of Bi2Te3 nanosheets with 1g PVP (a); Particle size distribution statistics of Bi2Te3 nanosheets (b); Surface and cross-section of PEDOT@Bi2Te3 composited film (c, d)
- To calculate ZT it is necessary to know thermal conductivity coefficient k. It is not clear how k was measured or evaluated (lines 412-414). k value is desirable.
- Thank you for your affirmative attitude and valuable comments. ZT is negatively correlated with thermal conductivity. According to the available literature, the intrinsic thermal conductivity of PEDOT is only 0.028-0.48 W/mK, which is much less than 1. The introduction of bismuth telluride nanosheets facilitates phonon scattering, which reduces the thermal conductivity of the lattice occupying 90% of the thermal conductivity, which in combination with the low thermal conductivity of the polymers, results in an effective reduction of thermal conductivity of the composite film. The ZT was calculated from the reported thermal conductivity of 0.07 W/mK of the composite film, which was useful in determining the potential application of the PEDOT@Bi2Te3 composite film in the field of thermoelectric devices. Regarding this issue, we have modified and enriched in the manuscript as follows: As the particle size of inorganic fillers has a limited impact on thermal conductivity [25], the ZT of PB-10 is calculated based on the thermal conductivity 0.07 W/mK of organic-inorganic hybrid films and compared with the current free-standing composite thermoelectric films.

Reviewer 2 Report
Comments and Suggestions for Authors
This paper reports the preparation, characterization, and applications of free-standing films of PEDOT@Bi2Te3 prepared by the tape casting. They hold promise for flexible thermoelectric technology in self-powered sensing applications. Bi2Te3 nanosheets fabricated by solvothermal method are tightly connected with flat-arranged PEODT molecules, forming an S-Bi bonded interface in the composite materials. The resulting bandgap is reduced and compared with the PEDOT film, the mobility and carrier concentration of the composite are significantly increased at room temperature and increases the conductivity as well.
Low-dimensional nanosheets were synthesized using a solvothermal method. PEDOT was combined with it via an electrostatic self-assembly method, and the free-standing PEDOT@Bi2Te3 composite films were prepared using a cast-film-forming process. The nanosheets were characterized using SEM, TEM, XRD, Raman and FT-IR spectroscopy, and XPS. The electrical property and Seebeck coefficient were measured using a thermoelectric performance testing system. This flexible thermoelectric film with self-supporting properties can have potential applications in the field of self-powered sensing using human thermal energy.
Some of my thoughts on the manuscript are as follows:
1. The manuscript is technically sound and well written, although I felt the author jumps into the technical core too soon without providing much background on thermoelectric materials. I believe the journal “polymers” has a broad readership and it might be nice to provide some background knowledge to the wider readers.
2. In line with the above comment, some of the technical terms like “Seeback coefficient” and “Phonon Scattering” have been used vastly but they have not been given a good introduction in the text. It may be helpful for the readers if the authors start with a basic introduction to such terminologies and take things from there.
3. Several characterization techniques have been used in the work. More details on the sample prep and procedure of each technique might be helpful for fellow scientists in the field.
4. A vast portion of the results and discussion section talks about the results of the characterization tests which include lots of numbers and parameters. Can these results be put in some tabulated form for each test for better readability?
5. Regarding the properties, do the nanosheets have any preferred orientation of the crystals (because that might be correlated with the conductivity)? Can a technique such as grazing incidence XRD give some insights?
6. I also feel the text needs to talk about other self supporting films studied for thermoelectric properties and ow the current material compares to them.
7. This is a minor comment, but the title sounds a little generic (like title of a review paper). Can it be made more material or application specific?

Author Response
Responses to the Referees’ comments
The detailed corrections are listed below point by point:
Comments to the Author
This paper reports the preparation, characterization, and applications of free-standing films of PEDOT@Bi2Te3 prepared by the tape casting. They hold promise for flexible thermoelectric technology in self-powered sensing applications. Bi2Te3 nanosheets fabricated by solvothermal method are tightly connected with flat-arranged PEODT molecules, forming an S-Bi bonded interface in the composite materials. The resulting bandgap is reduced and compared with the PEDOT film, the mobility and carrier concentration of the composite are significantly increased at room temperature and increases the conductivity as well.
Low-dimensional nanosheets were synthesized using a solvothermal method. PEDOT was combined with it via an electrostatic self-assembly method, and the free-standing PEDOT@Bi2Te3 composite films were prepared using a cast-film-forming process. The nanosheets were characterized using SEM, TEM, XRD, Raman and FT-IR spectroscopy, and XPS. The electrical property and Seebeck coefficient were measured using a thermoelectric performance testing system. This flexible thermoelectric film with self-supporting properties can have potential applications in the field of self-powered sensing using human thermal energy.
Some of my thoughts on the manuscript are as follows:
- The manuscript is technically sound and well written, although I felt the author jumps into the technical core too soon without providing much background on thermoelectric materials. I believe the journal “polymers” has a broad readership and it might be nice to provide some background knowledge to the wider readers.
- Response: Thank you for your affirmative attitude and valuable comments. Thermoelectric material is an important functional material, it can be electrical energy and thermal energy conversion. Thermoelectric devices manufactured with its energy saving, non-polluting to the environment, good stability and long service life and many other advantages. Therefore, in the modern construction of environmentally friendly social applications are increasingly important. Regarding this issue, we have modified and enriched the introduction in the manuscript as follows: Thermoelectric material is an important functional material, it can be electrical energy and thermal energy conversion. Thermoelectric devices manufactured with its energy saving, non-polluting to the environment, good stability and long service life and many other advantages. Therefore, in the modern construction of environmentally friendly social applications are increasingly important. Harnessing waste heat to generate electrical energy is essential for the swift advancement of alternative energy technologies. Thermoelectric devices have the capability to directly transform waste heat into electrical energy [1-4].
- In line with the above comment, some of the technical terms like “Seeback coefficient” and “Phonon Scattering” have been used vastly but they have not been given a good introduction in the text. It may be helpful for the readers if the authors start with a basic introduction to such terminologies and take things from there.
- Response: Thanks for your suggestions. Seebeck coefficient (S) is a measure of the magnitude of the voltage induced by a temperature difference in a thermoelectric material. Conductivity (σ) is a parameter used to describe the ease of charge flow in a substance. Thermal conductivity (κ) is the amount of heat transferred per unit of time per unit of horizontal cross-sectional area. These three parameters interact with each other to affect the efficiency of thermoelectric conversion. Phonon scattering is the process of energy exchange between carriers and phonons (energy quanta of the lattice vibrational law) in crystals, which is temperature and carrier transport dependent. Regarding this issue, we have modified and enriched it in the manuscript.
- Several characterization techniques have been used in the work. More details on the sample prep and procedure of each technique might be helpful for fellow scientists in the field.
- Response: Thanks for your kind reminder. The surface and sectional morphology of Bi2Te3 powder and PEDOT@Bi2Te3 film was investigated by scanning electron microscopy (scanning electron microscopy). The Raman spectroscopy of PEDOT@Bi2Te3 film were acquired using Renishaw-InVia equipped with a 532 nm red laser and a CCD detector. Infrared spectra of the films were performed by a Fourier-transform infrared spectrometer (FT-IR; Vertex 70). The doping levels and element valence of the composite film were determined through X-ray photoelectron spectroscopy (AXIS SUPRA), and XPS spectra were calibrated with the binding energy of the standard C 1s (284.8 eV). The characteristic parameters of the nanofilms, including conductivity, Seebeck coefficient, and power factor were evaluated using a thermoelectric test system (CTA-3), and the conductivity and Seebeck coefficient were observed employing static direct current and four-terminal methods under a helium atmosphere. The flexibility and sensitivity of the composite films were measured by an Keithley 2400 SourceMeter instrument. Carrier concentration and mobility inside the self-supported PEDOT@Bi2Te3 flexible thermoelectric films were characterized using the MMR K2000 controlled and continuously variable temperature Hall effect test system from the U.S.A. The tested film was a 10 mm×10 mm square with a thickness of 10 μm, and a magnetic field strength of 5000 G was applied. Regarding this issue, we have modified and enriched the experimental section in the manuscript.
- A vast portion of the results and discussion section talks about the results of the characterization tests which include lots of numbers and parameters. Can these results be put in some tabulated form for each test for better readability?
- Response: Thank you very much for your valuable advice. We checked all of the data carefully again. The construction of two-dimensional binary composite films effectively enhanced the thermoelectric properties, and the Seebeck coefficient and conductivity were positively correlated with the power factor of the composite films. Carrier concentration is positively correlated with conductivity and negatively correlated with Seebeck coefficient, so the relationship between carrier concentration and conductivity and Seebeck coefficient is illustrated in Table 1. In addition, defect energy levels generated in the binary composite films decouple the conductivity from the Seebeck coefficient, achieving enhanced thermoelectric properties. Regarding the table, we have modified and enriched the results and discussion in the manuscript.
Table 1 Comparison of thermoelectric parameters for a series of composite films
- Regarding the properties, do the nanosheets have any preferred orientation of the crystals (because that might be correlated with the conductivity)? Can a technique such as grazing incidence XRD give some insights?
- Response: Thank you very much for your valuable advice. Bi2Te3 belongs to the layered hexagonal crystal system with R-3m space group symmetry, and this layered structure provides the basis for its good thermoelectric properties. At the nanoscale, it is also theoretically possible to form Bi2Te3 nanosheets with selective orientation if the growth direction and lamellar structure of the nanosheets can be controlled by suitable preparation methods. In this study, Bi2Te3 nanosheets were prepared by solvothermal method. Solvothermal method can successfully synthesized Bi2Te3 nanosheets originated from the suitable addition of PVP. As TeO2 reacted with NaOH to produce TeO32-, the first production between TeO32- under pyrolysis conditions is Te monomer, while PVP K30, as an organic substance with a molecular weight (40000) larger than that of Bi2Te3, will produce a spatial site-blocking effect, which will inhibit the too-fast reaction of TeO32- and TeO32- to combine to produce Te monomers. Therefore, when the amount of PVP is not appropriate, it would still lead to the generation of excessive Te monomers. In contrast, when the addition of PVP was 1 g, Bi2Te3 nanosheets synthesized by the solvothermal method had a complete hexagonal morphology and a lamellar structure with a thickness of 20 nm. Because Bi2Te3 is a rhombohedral hexahedral lamellar structure, each atomic layer is arranged along the C-axis in the manner of Te-Bi-Te, the two neighboring Te layers are bonded by van der Waals force, and the atoms within the layer are bonded by atomic bonds, and the chemical force between the atomic bonds is much larger than the van der Waals force. Bi2Te3 is more likely to grow along the lamellar growth of the bonding force is large, and the naturally growing Bi2Te3 is easy to be in the form of lamellae.
- I also feel the text needs to talk about other self-supporting films studied for thermoelectric properties and the current material compares to them.
- Response: Thank you very much for your affirmative attitude and valuable comments. We discussed the thermoelectric properties of flexible free-standing films based on PEDOT-Bi2Te3 as shown in Figure 10d. The ZT value of the composite film prepared by organic-inorganic hybridization is higher than that of the layered-coated composite films. Zheng et al. [8] introduced Bi2Te3 nanowires, enhancing both conductivity and Seebeck coefficient is achieved. However, films prepared using the spin-coating method lack independent self-supporting properties. In this study, the performance of PB-10 prepared by the flow casting of Bi2Te3 nanosheets into PEDOT is second only to it. Further exploration of the process and performance improvement mechanisms is needed to enhance the thermoelectric performance of composite films.
Figure 10 Temperature dependence of PEDOT@Bi2Te3 nanofilms: conductivity (a); Seebeck coefficient (b); Power factor (c); Comparison of ZT for the free-standing composited films (d).
- This is a minor comment, but the title sounds a little generic (like title of a review paper). Can it be made more material or application specific?
- Response: Thank you very much for your valuable and constructive advice. The title is central to clearly expressing the content of the study. A title like “One Stone, three birds: Feasible tuning of barrier heights induced by hybridized interface in free-standing PEDOT@Bi2Te3 thermoelectric films” would be more informative for a wide audience. I have changed the title in the main text.

Round 2
Reviewer 1 Report
Comments and Suggestions for Authors
The content of the file with author’s response to reviewer indicates that authors planned to make some changes in the manuscript. Surprisingly, only two of them (marked yellow) are reflected in the revised manuscript, namely #1(Introduction) and #2(Conclussion). A careful word-by-word comparison of the responses and the updated text of the manuscript made it possible to say that responses for comments #3-6,9,10 (marked red) were formulated but not included in the text. Thus, the final version is almost the same as the previous one.

Author Response
Thank you very much for your reminder. I have corrected and yellow-marked all response comments in the revised manuscript. Please check it out.

Round 3
Reviewer 1 Report
Comments and Suggestions for Authors
The authors made some planned improvements to the text of the article and marked the corrected parts with colored highlighting. This type of submission of a revised manuscript is generally accepted, convenient for its re-review and, if the authors were able to notice, speeds up the review process. The current version, despite some shortcomings, can be recommended for publication.